# Synthesis, APPI Mass-Spectrometric Characterization, and Polymerization Studies of Group 4 Dinuclear Bis(*ansa*-metallocene) Complexes

**Gilles Schnee [1], Mathilde Farenc [2,3,4] , Leslie Bitard [1] , Aurelien Vantomme [5], Alexandre Welle [5], Jean-Michel Brusson [6], Carlos Afonso [2,4] , Pierre Giusti [3,4], Jean-François Carpentier [1,* and Evgueni Kirillov [1,***

[1]  Univ Rennes, CNRS, ISCR (Institut des Sciences Chimiques de Rennes), UMR 6226, F-35000 Rennes, France;
    gilles.schnee@wanadoo.fr (G.S.); leslie.bitard@etudiant.univ-rennes1.fr (L.B.)
[2]  CNRS, COBRA, Univ Rouen, UMR 6014, F-76821 Mont Saint Aignan, France;
    mathildefarenc@gmail.com (M.F.); carlos.afonso@univ-rouen.fr (C.A.)
[3]  Total Research and Technologies Gonfreville, BP 27, F-76700 Harfleur, France; pierre.giusti@total.com
[4]  Total Raffinage Chimie, Joint Laboratory C2MC (Complex Matrices Molecular Characterization), Univ Pau,
    Univ Rouen, CNRS, F-64053 Pau, France
[5]  Total Research and Technologies Feluy, Zone Industrielle C, B-7181 Feluy, Belgium;
    aurelien.vantomme@total.com (A.V.); alexandre.welle@total.com (A.W.)
[6]  Total SA, Direction scientifique, F-92069 Paris La Défense, France; jean-michel.brusson@total.com
*  Correspondence: jean-francois.carpentier@univ-rennes1.fr (J.-F.C.); evgueni.kirillov@univ-rennes1.fr (E.K.);
    Tel.: +33-223-236-118 (E.K.)

**Abstract:** New ligand platforms of the type *p*- or *m*-Ph{-CR(3,6-*t*Bu$_2$Flu)(Cp)}$_2$ (*para*-, R = Me (**2a**), H (**2b**); *meta*-, R = Me (**2c**)) were synthesized via nucleophilic addition of the 3,6-*t*Bu$_2$-fluorenyl-anion onto the parent phenylene-bridged difulvenes (**1a–c**). The corresponding discrete homodinuclear zirconium and hafnium bis(dichloro *ansa*-metallocene) complexes, Ph[{-CR(3,6-*t*Bu$_2$Flu)(Cp)}MCl$_2$]$_2$ (*p*-, R = Me (**3a-Zr$_2$**, **3a-Hf$_2$**), R = H (**3b-Zr$_2$**); *m*-, R = Me (**3c-Zr$_2$**), were prepared by salt metathesis reactions. An attempt to generate in situ a heterodinuclear complex **3a-Zr-Hf** was also undertaken. For the first time, Atmospheric Pressure PhotoIonization (APPI) mass-spectrometric data were obtained for all dinuclear compounds and found to be in excellent agreement with the simulated ones. Preliminary studies on the catalytic performances of these dinuclear complexes, upon activation with MAO, in ethylene homopolymerization and ethylene/1-hexene copolymerization revealed a few differences as compared to those of the monometallic analogues. In particular, slightly lower molecular weights and a greater formation of short methyl and ethyl branches were obtained with the dinuclear systems.

**Keywords:** metallocene catalysts; mass-spectroscopy; olefin polymerization

## 1. Introduction

The development and applications of multinuclear group 4 metal $\alpha$-olefin polymerization catalysts have increased dramatically in the past decade [1,2]. The interest in this area has primarily been driven by the potential to exploit intermetallic cooperativity/synergism between the two or more proximal metal centers to eventually enhance the performance of polymerization systems. For instance, several studies on dinuclear catalysts suggest that catalyst activity [3–13] and molecular weight, [3,7,10,14] tacticity, [13–15], or comonomer incorporation [8,16–20] of/in the resulting polymers

can be greater than those of the corresponding mononuclear analogues that have isostructural catalytic sites.

Group 4 metal catalysts based on one-carbon-bridged *ansa*-cyclopentadienyl-fluorenyl platforms, {R$_2$C(Cp/Flu)}$^{2-}$, hold a unique position in $\alpha$-olefin polymerization thanks to their high catalytic activity, excellent control, and remarkable stereospecificity [21–33]. However, only a few examples of dinuclear systems of this type have been reported in the literature (Scheme 1) [9,34–37].

**Scheme 1.** Examples of group 4 dinuclear bis(dichloro *ansa*-metallocene) complexes that incorporate {R$_2$C(Cp/Flu)} ligand platforms [8,9].

We herein report on the synthesis of dinuclear group 4 bis(Cp/Flu-metallocene) complexes, linked at the C1-bridge by a *para*- or *meta*-phenylene moiety [3,18–20,38], as well as their characterization by NMR spectroscopy and advanced Atmospheric Pressure PhotoIonization (APPI) mass-spectrometric methods. The catalytic performances of the synthesized complexes, after their activation with MAO, were preliminarily investigated in homogeneous ethylene homopolymerization and ethylene/1-hexene copolymerization, and compared to those of the mononuclear *ansa*-metallocene complexes as a reference.

## 2. Results and Discussion

### 2.1. Synthesis of Proligands

An efficient and scalable synthesis via nucleophilic addition of the (3,6-di-*tert*-butyl)fluorenyl anion onto fulvenes is regularly utilized to prepare one-carbon-bridged R$_2$C{Cp/Flu}H$_2$ proligands [21–28]. This methodology was here extended to three different bis(fulvene) platforms (**1a–c**), which were prepared from cyclopentadiene and the corresponding aromatic diketones or dialdehydes (Scheme 2), and isolated in good yields as yellow or orange solids.

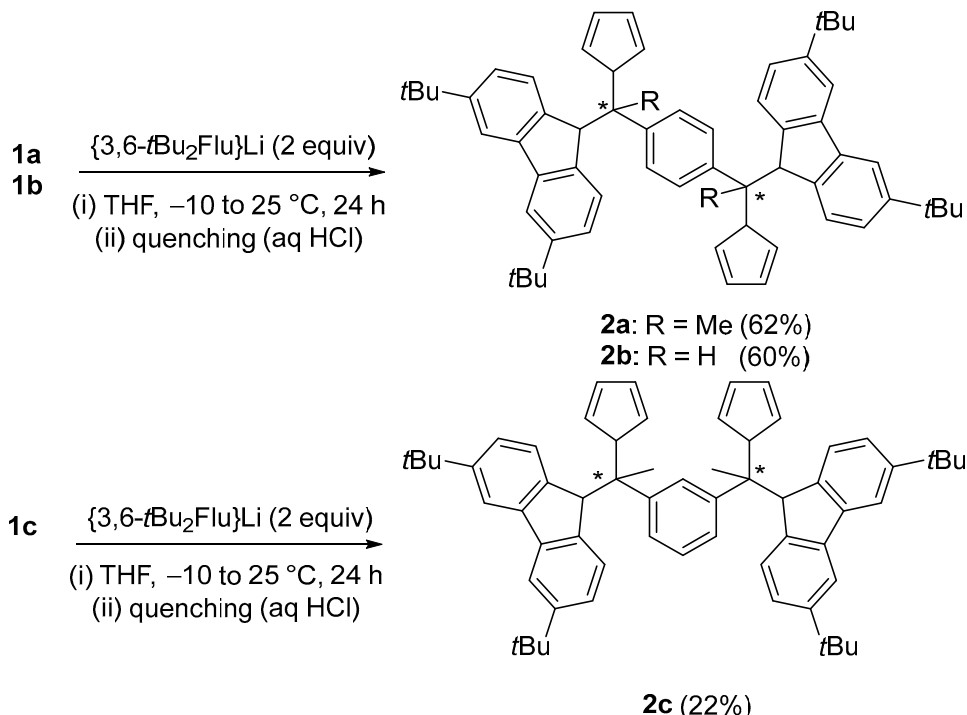

**Scheme 2.** Synthesis of the *para*- and *meta*-phenylene-bridged bis(fulvenes) **1a–c**.

To prepare the targeted *p*- and *m*-Ph{-CR(3,6-*t*Bu₂FluH)(CpH)}₂ proligands **2a–c**, bis(fulvenes) **1a–c** were subsequently reacted with two equiv. of the {3,6-*t*Bu₂Flu}⁻ anion in THF (Scheme 3). After workup, the corresponding *para*-bridged proligands **2a,b** were isolated in good yields; synthesis/recovery of the *meta*-bridged proligand **2c** proved somehow to be more difficult (22% isolated yield). This lower yield may be due to a larger steric hindrance in the final *meta*-phenylene-bridged product imposed by the two very bulky {Cp/Flu} moieties as compared to the *para* analogues.

**Scheme 3.** Synthesis of the proligands **2a–c**.

These proligands are stable at room temperature in solution and in the solid state, and their structures were authenticated by ¹H and ¹³C NMR spectroscopy (Figures S9, S10, S12, S13, S15,

and S16) and ESI-mass spectrometry (Figures S11, S14, and S17). The NMR data for these compounds appeared to be complicated by the presence of two stereogenic centers in the molecules, resulting in the existence of pairs of diastereoisomers in each case, and also of their different tautomers (i.e., isomers of C=C bonds within the CpH rings). The ESI-MS measurements data showed clearly the expected molecular $[M + H]^+$ ions at $m/z$ 815.55, 787.52 for **2a,b** and at 853.51 ($[M + K]^+$) for **2c**.

## 2.2. Synthesis of Group 4 Dinuclear Bis(dichloro ansa-metallocene) Complexes

In order to prepare the corresponding group 4 bis(dichloro *ansa*-metallocene) complexes, standard salt-metathesis reactions between the ligand tetraanions, generated in situ in Et$_2$O, and MCl$_4$ salts (2 equiv.), were used (Scheme 4). Thus, the homodinuclear bis(dichloro *ansa*-zirconocenes) **3a-c-Zr$_2$** and bis(dichloro *ansa*-hafnocene) **3a-Hf$_2$** were isolated in good yields as red and yellow solids, respectively.

**Scheme 4.** Synthesis of the group 4 homodinuclear bis(dichloro *ansa*-metallocene) complexes.

As **3a-c-Zr$_2$** and **3a-Hf$_2$** were derived from diastereomeric mixtures of proligands, two diastereomers for each of these compounds were anticipated, featuring $C_s$-/$C_i$-symmetries for the *para*-phenylene-bridged complexes **3a,b-M$_2$** and $C_s$-/$C_1$-symmetries for the *meta*-phenylene-bridged complex **3c-Zr$_2$** (Scheme 5). Accordingly, the $^1$H and $^{13}$C NMR spectra of the crude **3a-Zr$_2$** (Figures S24–S26), **3a-Hf$_2$** (Figures S28 and S29, respectively), and **3c-Zr$_2$** (Figures S35 and S36, respectively) complexes displayed two sets of resonances corresponding to the two diastereomers. Unexpectedly, only one set of resonances assigned to a single diastereoisomer of either $C_s$- or $C_i$-symmetry was observed in the $^1$H NMR spectrum of **3b-Zr$_2$** (Figure S31). As this compound was isolated in a lower yield than the other ones, one cannot discard that only one diastereoisomer was recovered in the workup.

**Scheme 5.** Possible diastereoisomers of the phenylene-bridged bis(dichloro *ansa*-metallocene) complexes.

Unfortunately, all attempts to grow single-crystals of these complexes suitable for X-ray diffraction studies have failed so far. However, the identity of these bis(dichloro *ansa*-metallocene) compounds was confirmed unambiguously by mass spectrometry (vide infra).

In order to obtain a better clue about the possible structures of the dinuclear bis(metallocenes), the corresponding geometries of the two $C_s$- and $C_i$-symmetric isomers of **3a-Zr$_2$** (Figure S68; see the Experimental Section for details) and the two $C_s$- and $C_1$-symmetric isomers of **3c-Zr$_2$** (Figure S69) were modeled by DFT computations. It is noteworthy that the optimized geometries of the isomers belonging to both dinuclear systems **3a-Zr$_2$** and **3c-Zr$_2$** featured relatively long Zr . . . Zr intermetallic distances of 10.5–10.8 Å and 9.2–9.8 Å, respectively. Also, the respective orientations of the metallocenic fragments in these structures resulted in the coordination sites, represented by the chlorine ligands, pointing in opposite directions. Such an orientation of the metallocenic moieties in both *para*- and *meta*-phenylene-bridged systems may not be favorable to the mutual approach of the two metal centers in dinuclear active species derived thereof during polymerization (vide infra). Note, however, that the above observations were made on the most stable *neutral* isomers as determined by DFT, and they do not necessarily reflect the proximity that can be reached from dynamic conformations in those species. Also, the behavior of the active *cationic* species associated with counterionic moieties may be quite different.

In an attempt to synthesize a hetero-bis(metallocene) incorporating both zirconium and hafnium metals, a similar salt-metathesis protocol as that utilized for the synthesis of the homo-bis(dichloro *ansa*-metallocenes) **3a-Zr$_2$** and **3a-Hf$_2$** was probed using 1 equiv. of each of the metal precursors ZrCl$_4$ and HfCl$_4$ (Scheme 6). In this case, as anticipated, a statistical 1:1:2 mixture of the homodinuclear **3a-Zr$_2$** and **3a-Hf$_2$** complexes and the heterodinuclear **3a-Zr/Hf** complex was obtained, as revealed by [1]H NMR spectroscopy of the crude sample. No single-crystal suitable for X-ray diffraction studies has been grown thus far. Due to the complexity of the mixture and the obvious difficulties associated with regular elemental and spectroscopic analyses, only its mass-spectrometric characterization was performed (vide infra).

**Scheme 6.** Attempted synthesis of the heterodinuclear bis(metallocene) **3a-Zr/Hf**.

### 2.3. Synthesis of Mononuclear Ansa-metallocene Analogues

For comparative studies of mass-spectrometric analyses and of catalytic properties of the bis(metallocene)s in α-olefin polymerization, their mononuclear analogues were also synthesized (Scheme 7). The complexes **3a',b'-Zr** and **3b'-Hf** were isolated in good yields and characterized by $^1$H and $^{13}$C NMR spectroscopic studies, X-ray diffraction (for **3a'-Zr**; Figure S67), and APPI mass-spectrometry (vide infra).

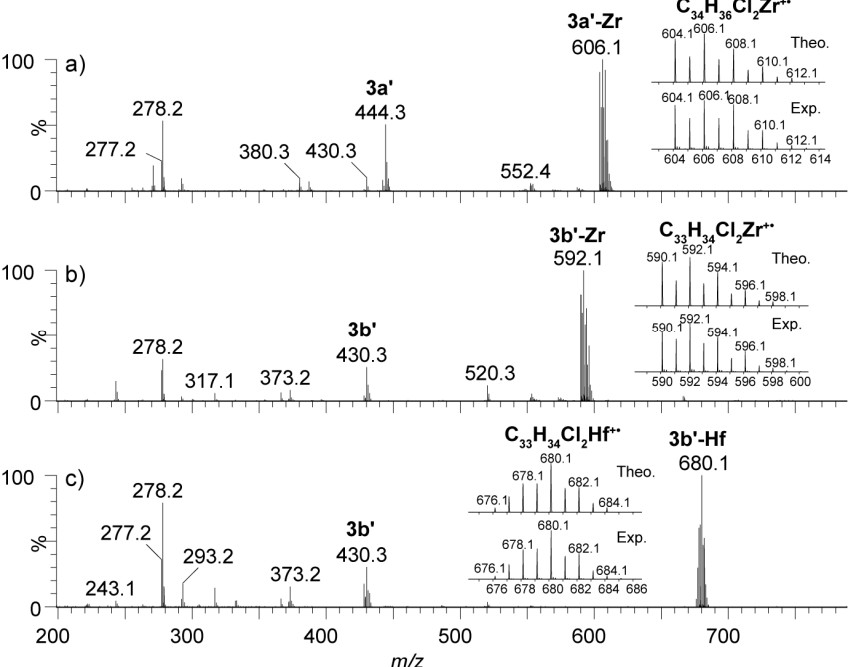

**Scheme 7.** Synthesis of the mononuclear metallocene analogues.

### 2.4. Mass Spectrometric Studies of Mononuclear and Dinuclear Bis(dichloro ansa-metallocene) Complexes

Atmospheric Pressure PhotoIonization (APPI) was chosen instead of the more common electrospray (ESI), as APPI is very efficient for the ionization of aromatic molecules that do not contain polar groups; also, it allows for the use of dry toluene as a solvent to preserve the rather sensitive metallocene complexes [39,40]. The APPI mass spectra of the mononuclear complexes **3a'-Zr**, **3b'-Zr**, and **3b'-Hf** are summarized in Figure 1. In each case, the corresponding intact species was detected as a M$^{+\bullet}$ molecular ion. A free ligand was also observed in the spectra at $m/z$ 444.3 and $m/z$ 430.3, as well as the $C_{21}H_{26}$ moiety at $m/z$ 278.

**Figure 1.** The APPI(+) mass spectra of the mononuclear complexes **3a'-Zr** (**a**), **3b'-Zr** (**b**), and **3b'-Hf** (**c**). The zoomed areas showcase the theoretical and experimental isotopic clusters. For each isotopic distribution, only the most intense isotope peak is labelled.

By analogy, the $M^{+\bullet}$ molecular ions were identified for the dinuclear bis(zirconocene) **3a-b-Zr$_2$** compound and the dinuclear bis(hafnocene) **3a-Hf$_2$** compound (Figure 2). The accurate masses and isotopic distributions ($m/z$ 1130.2074, 1102.1754, and 1310.2865, respectively) are in very good agreement (<5 ppm) with those expected theoretically based on the corresponding ions' molecular formula ($m/z$ 1130.2013, 1102.1700, and 1310.2850, respectively). In addition to the dinuclear species **3a,b-Zr$_2$** and **3a-Hf$_2$**, molecular ions derived from the monometallated fragments, i.e., **3a,b-Zr** and **3a-Hf**, were detected in each case at $m/z$ 972.3776, 944.3398, and 1062.4139, respectively. These ions were most likely not generated by gas-phase fragmentation, as they were not produced by collision-induced dissociation of the dinuclear molecular ions. They, however, may have been produced by partial degradation (e.g., hydrolysis) during the sample handling or in the atmosphere source that may contain traces of water.

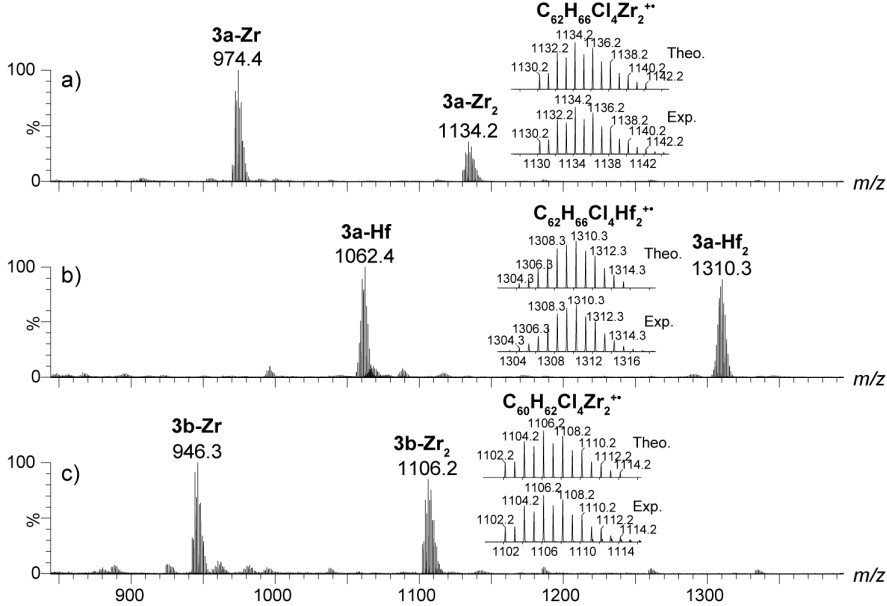

**Figure 2.** The APPI(+) mass spectra of the dinuclear bis(metallocene) **3a-Zr$_2$** (**a**), **3a-Hf$_2$** (**b**), and **3b-Zr$_2$** (**c**) complexes. The zoomed areas showcase the theoretical and experimental isotopic clusters. For each isotopic distribution, only the most intense isotope peak is labelled.

For the mixture containing the heterodinuclear bis(dichloro *ansa*-metallocene) compound **3a-Zr/Hf** and its homodinuclear counterparts **3a-Zr$_2$** and **3a-Hf$_2$**, a set of five isotopic distributions was observed in the APPI mass-spectrum (Figure 3). Besides the distributions corresponding to the homodinuclear **3a-Zr$_2$** and **3a-Hf$_2$** and their monometallated versions, i.e., **3a-Zr** and **3a-Hf**, respectively, an isotopic distribution at $m/z$ 1222.2463 was identified and unequivocally assigned to **3a-Zr/Hf**. APPI should present a low ionization discrimination for these species, so their relative abundance should be representative of their actual amount in the sample, although the most air-sensitive molecules may present a lower abundance because of higher degradation. This possibly accounts for the observed lower intensity of peaks arising from **3a-Hf$_2$**.

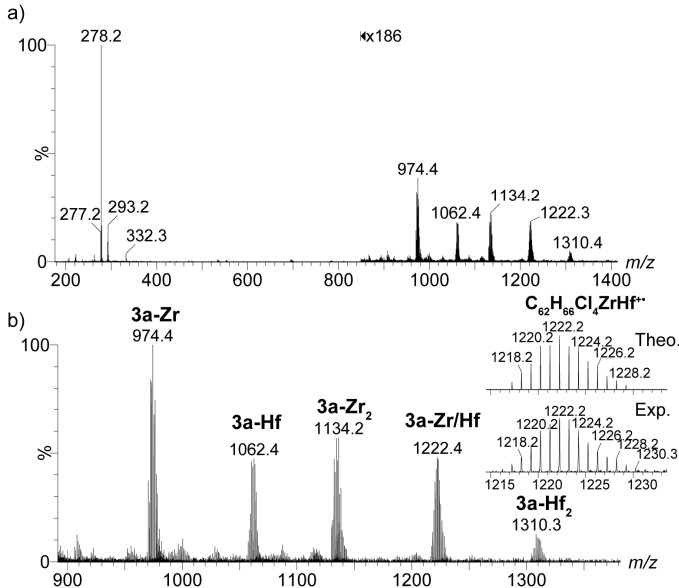

**Figure 3.** (**a**) The APPI(+) mass spectrum of the crude reaction mixture resulting from metallation of the proligand **3a** by a 1:1 mixture of ZrCl$_4$ and HfCl$_4$; (**b**) The enlargement of the $m/z$ 900–1350 area. The zoomed areas showcase the theoretical and experimental isotopic clusters of the dinuclear bis(dichloro *ansa*-metallocene) complexes **3a-Zr-Hf**. For each isotopic distribution, only the most intense isotope peak is labelled.

## 2.5. Polymerization Catalysis

The dinuclear bis(dichloro *ansa*-metallocene) complexes **3a-M$_2$**, **3b-Zr$_2$**, and **3c-Zr$_2$**, and their mononuclear analogues **3a,b'-Zr** and **3b'-Hf**, in combination with methylalumoxane (MAO), were evaluated in homogeneous ethylene polymerization (Table 1) and ethylene/1-hexene copolymerization (Table 2). Each polymerization experiment was repeated independently twice under the same conditions (toluene, 5.5 bar of ethylene, 60 °C), which revealed good reproducibility in terms of productivity (polymer yield) and physicochemical properties ($T_m$, $M_w$, polydispersity index (PDI)) of the isolated polymer.

**Table 1.** Homopolymerization of ethylene [a].

| Entry | Comp. | $m_{PE}$ (g) | Prod. (kg·mol$^{-1}$·h$^{-1}$) | $M_w$ [b] (kg·mol$^{-1}$) | $M_w/M_n$ [b] | $T_m$ [c] (°C) | %Me [d] (wt%) | %Et [d] (wt%) | %$n$Bu [d] (wt%) |
|---|---|---|---|---|---|---|---|---|---|
| 1 | **3a-Zr$_2$** | 6.20 | 24,800 | 175.1 | 3.2 | 132.3 | 0.0 | 0.1 | 0.0 |
| 2 | **3a-Zr$_2$** | 5.62 | 22,500 | 196.6 | 3.4 | 132.2 | 0.0 | 0.2 | 0.0 |
| 3 | **3a'-Zr** | 5.64 | 22,600 | 260.4 | 3.6 | 132.1 | 0.0 | 0.1 | 0.0 |
| 4 | **3a'-Zr** | 6.14 | 24,600 | 307.9 | 4.1 | 131.8 | 0.0 | 0.0 | 0.0 |
| 5 | **3b-Zr$_2$** | 5.82 | 23,280 | 134.6 | 3.1 | 127.2 | 1.5 | 0.2 | 0.0 |
| 6 | **3b-Zr$_2$** | 5.20 | 20,800 | 189.2 | 3.5 | 129.0 | 0.9 | 0.2 | 0.0 |
| 7 | **3b'-Zr** | 6.79 | 27,160 | 180.9 | 3.4 | 132.1 | 0.0 | 0.1 | 0.0 |
| 8 | **3b'-Zr** | 5.27 | 21,080 | 272.6 | 4.3 | 133.2 | 0.0 | 0.1 | 0.0 |
| 9 | **3c-Zr$_2$** | 4.96 | 19,800 | 85.0 | 3.3 | nd | 0.0 | 0.2 | 0.0 |
| 10 | **3a-Hf$_2$** | 1.53 | 6100 | –[f] | –[f] | 132.7 | 0.0 | 0.0 | 0.0 |
| 11 [e] | **3a-Hf$_2$** | 1.63 | 6500 | –[f] | –[f] | 133.6 | 0.0 | 0.0 | 0.0 |
| 12 | **3b'-Hf** | 1.38 | 5500 | 1146.3 | 4,1 | nd | 0.0 | 0.0 | 0.0 |

[a] General conditions: toluene (100 mL), $T_{pol}$ = 60 °C, [Zr]$_0$ = 10.1 µmol·L$^{-1}$, $P_{ethylene}$ = 5.5 bar, [Al]$_0$/[Zr]$_0$ = 1000, polymerization time = 15 min, nd = not determined; [b] Determined by SEC at 135 °C in 1,2,4-trichlorobenzene; [c] Determined by DSC from a second heating run; [d] Determined by $^{13}$C NMR spectroscopy; [e] BHT (300 equiv. versus [Hf] was added to scavenge M$_3$Al of MAO); [f] Insoluble polymer was recovered, preventing an SEC analysis.

**Table 2.** Ethylene/1-hexene copolymerization [a].

| Entry | Comp. | $m_{PE}$ (g) | Prod. (kg·mol$^{-1}$·h$^{-1}$) | $M_w$ [b] (kg·mol$^{-1}$) | $M_w/M_n$ [b] | $T_m$ [c] (°C) | %Me [d] (wt%) | %Et [d] (wt%) | %$n$Bu [d] (wt%) |
|---|---|---|---|---|---|---|---|---|---|
| 1 | **3a-Zr₂** | 6.20 | 24,800 | 70.6 | 2.6 | 122.5 | 0.0 | 0.1 | 15.7 |
| 2 | **3a-Zr₂** | 5.62 | 22,500 | 84.3 | 3.0 | 123.8 | 0.0 | 0.0 | 23.2 |
| 3 | **3a′-Zr** | 7.22 | 28,900 | 129.3 | 3.4 | 117.7 | 0.0 | 0.0 | 22.6 |
| 4 | **3a′-Zr** | 6.77 | 27,100 | 132.1 | 3.3 | 118.2 | 0.0 | 0.0 | 21.3 |
| 5 | **3b-Zr₂** | 7.74 | 31,000 | 75.8 | 2.8 | 111.4 | 0.9 | 0.1 | 22.5 |
| 6 | **3b-Zr₂** | 6.70 | 26,800 | 64.0 | 2.7 | 112.2 | 0.4 | 0.1 | 19.4 |
| 7 | **3b′-Zr** | 7.57 | 30,300 | 89.9 | 2.7 | 113.9 | 0.0 | 0.0 | 21.1 |
| 8 | **3b′-Zr** | 7.30 | 29,200 | 96.9 | 2.9 | 108.2 | 0.0 | 0.0 | 22.7 |
| 9 | **3c-Zr₂** | 7.75 | 31,000 | 190.6 | 3.5 | 116.0 | 0.0 | 0.0 | 21.3 |
| 10 | **3a-Hf₂** | 3.78 | 15,100 | 447.1 | 3.1 | nd | 0.0 | 0.0 | 28.1 |
| 11 [e] | **3a-Hf₂** | 4.34 | 17,400 | 581.5 | 4.0 | nd | 0.0 | 0.0 | 29.6 |
| 12 | **3b′-Hf** | 1.66 | 6600 | -[f] | -[f] | nd | 0.0 | 0.0 | 28.5 |

[a] General conditions: toluene (100 mL), $T_{pol}$ = 60 °C, [Zr] = 10.1 μmol·L$^{-1}$, $P_{ethylene}$ = 5.5 bar, [Al]/[Zr] = 1000, [1-hexene]₀ = 0.2 M, polymerization time = 15 min, nd = not determined; [b] Determined by size-exclusion chromatography (SEC) at 135 °C in 1,2,4-trichlorobenzene; [c] Determined by DSC from a second run; [d] Determined by $^{13}$C NMR spectroscopy; [e] BHT (300 equiv. versus Hf) was added to scavenge M₃Al of MAO; [f] Insoluble polymer was recovered, preventing an SEC analysis.

In general, no significant or limited difference was observed for the experiments involving dinuclear bis(metallocene) with respect to those using the corresponding mononuclear analogues (Table 1; compare entries 1–2/3–4, 5–6/7–8, and 10–11/12, respectively). Indeed, all of the systems that were produced with similar productivities used polyethylene samples exhibiting quite similar molecular weight distributions and $T_m$ values. A slight drop in productivity was observed for the *meta*-bridged system **3c-Zr₂** (entry 9), also yielding a lower molecular weight polyethylene (PE) as compared to the *para*-bridged congeners **3a,b-Zr₂**. Interestingly, as established by $^{13}$C NMR spectroscopy, **3b-Zr₂** induced the formation of short branches (methyl and, to a lesser extent, ethyl) to a significantly greater extent than its mononuclear analogue and any other dinuclear system; the same observation was made in ethylene/1-hexene copolymerization (vide infra). This may presumably arise from a chain-walking mechanism.

The Hf-based systems **3a-Hf₂** and **3b′-Hf** appeared to be ca. 3-fold less productive (entries 10–12) than their Zr-based counterparts while affording much higher molecular weight PEs. The addition of BHT (entry 11) to scavenge the excess "free" AlMe₃ [41] present in MAO did not affect the productivity of this system, and also resulted in an insoluble polymer (likely due to the formation of high molecular weight polyethylene (HMWPE) because of the absence of transfer to AlMe₃).

Also, no strikingly different results were obtained upon using the different mono- and dinuclear compounds in ethylene/1-hexene copolymerization (Table 2). The Zr-based systems, both di- and mononuclear, afforded a very narrow range of productivities. It is of note that, in the *para*-bridged series **3a,b-Zr₂** and **3a′b′-Zr**, the dinuclear bis(metallocenes) gave slightly but significantly lower molecular weight copolymers and narrower dispersities (compare entries 1–2/3–4 and 5–6/7–8, respectively) than their respective mononuclear analogues. Conversely, the *meta*-bridged system **3c-Zr₂** afforded a higher molecular weight copolymer than its mononuclear analogue (compare entries 3/4 and 9). Again, higher molecular weight copolymers were obtained with Hf-based catalysts, but with lower productivities than their Zr counterparts (entries 10–12). Also, a somewhat higher incorporation of 1-hexene in copolymers was achieved with the Hf-based catalysts.

## 3. Materials and Methods

### 3.1. General Considerations

All experiments were performed under a dry argon atmosphere, using a glovebox or standard Schlenk techniques. THF and Et₂O were distilled prior to use from sodium benzophenone ketyl. Hexane and heptane were distilled from CaH₂ and stored over 3 Å molecular sieves. Deuterated solvents (benzene-$d_6$, toluene-$d_8$ > 99.5% D; Euriso-top, Saint-Aubin, France) were

distilled from Na/K alloy or $CaH_2$ (for $CD_2Cl_2$) and stored over 3 Å molecular sieves. $CDCl_3$ (99.8% D, Euriso-top) was used as received. Cyclopentadiene (Acros, Geel, Belgium) was freshly distilled prior to use. 3,6-Di-*tert*-butyl-fluorene and MAO were generously provided by Total Raffinage-Chimie. The fulvene precursors, (1-(cyclopenta-2,4-dien-1-ylidene)ethyl)benzene and (cyclopenta-2,4-dien-1-ylidenemethyl)benzene [42], were prepared following protocols published in the literature. $ZrCl_4$ and $HfCl_4$ were used as received (anhydrous, Strem Chemicals, Bischheim, France). 1-Hexene (Fisher Chemical, Illkirch, France) was distilled and stored over 3 Å molecular sieves under argon. Ethylene (N35, Air Liquide, Paris, France) was used without further purification.

### 3.2. Instruments and Measurements

The NMR spectra of air- and moisture-sensitive compounds were recorded on Bruker AM-400 and AM-500 spectrometers in Teflon-valved NMR tubes at room temperature. $^1$H and $^{13}$C chemical shifts are reported in ppm versus $SiMe_4$ and were determined using residual solvent signal. Coupling constants are given in Hertz. Assignments of signals were carried out using 1D ($^1$H, $^{13}$C{$^1$H}, JMOD) and 2D (COSY, HMBC, HMQC) NMR experiments. Elemental analyses were performed on a Carlo Erba 1108 Elemental Analyzer instrument at the London Metropolitan University by Stephen Boyer or on a Flash EA1112 CHNS Thermo Electron apparatus at CRMPO, Rennes, and were the average of a minimum of two independent measurements.

The $^{13}$C{$^1$H} NMR and GPC analyses of polymer samples were performed in the research center of Total Research and Technologies in Feluy (Feluy, Belgium). The $^{13}$C{$^1$H} NMR analyses were run on a 500 MHz Bruker Avance III with a 10 mm cryoprobe HTDUL in trichlorobenzene/$C_6D_6$ (2 mL/0.5 mL). The GPC analyses were performed in 1,2,4-trichlorobenzene at 135 °C using PS standards for calibration. Differential scanning calorimetry (DSC) analyses were performed on a Setaram DSC 131 apparatus under a continuous flow of helium and using aluminum capsules. Glass transition and melting temperatures were measured during the second heating (10 °C·min$^{-1}$).

The ESI (ElectroSpray Ionization) mass spectra of organic compounds, including proligands, were recorded at CRMPO-Scanmat (Rennes, France) on an Orbitrap Thermo Fisher Scientific Q-Exactive instrument with an ESI source in positive mode by direct introduction at 5–10 µg·mL$^{-1}$. Samples were prepared in $CH_2Cl_2$ at 10 µg·mL$^{-1}$.

The ASAP (Atmospheric Solids Analysis Probe) mass spectra of proligands were recorded at the CRMPO-Scanmat (Rennes, France) on a Q-TOF Bruker Maxis 4G instrument with an APCI source in positive mode at desorption temperatures of 255 and 300 °C.

### 3.3. APPI Mass Spectrometric Characterization of Metal Complexes

The mass spectra of metal complexes were recorded on a hybrid quadrupole time-of-flight instrument (Waters, Synapt G2, Manchester, England) equipped with an APPI source. The instrument was operated in the positive ion mode. The ionization experimental conditions were set as follows: desolvation gas flow, 700 L h$^{-1}$; source temperature, 120 °C; probe temperature, 400 °C; sampling cone, 20 V; extraction cone, 3 V. The time-of-flight was operated in the 'resolution mode' yielding a resolving power of about 20,000. The samples were prepared in a glove box using dried toluene in 1.5 mL glass vials with a final concentration of 20 µM. For analysis, the sample was taken with a dry syringe stored in an oven. The solution was directly infused into the source using a syringe pump at a flow rate of 200 µL h$^{-1}$. Data were acquired over the *m/z* 50–2000 range for 2–5 min. Note that the given accurate masses are given with Water Mass Lynx 4.1 that do not take into account the mass of the electron removed during ionization. All given masses are monoisotopic values. In the mass spectra, only the highest abundant isotope is labelled.

### 3.4. 1,4-Bis(1-(cyclopenta-2,4-dien-1-ylidene)ethyl)benzene (**1a**)

In a 250 mL round bottom flask equipped with a magnetic stirring bar and an argon inlet, freshly cracked cyclopentadiene (12.36 mL, 148.0 mmol) and 1,4-diacetylbenzene (4.82 g, 30.0 mmol)

were dissolved in methanol (200 mL). To this solution, pyrrolidine (7.5 mL, 89.0 mmol) was added at 0 °C. The reaction mixture was stirred at room temperature for 7 days. After neutralization with glacial acetic acid (7.5 mL) and separation of the organic phase, volatiles were evaporated under vacuum to give a yellow powder (5.51 g, 21.3 mmol, 72%). $^1$H NMR (CDCl$_3$, 400 MHz, 25 °C): δ 7.35 (s, 4H, C*H*-Ar), 6.59 (dt, $^3J$ = 5.2, $^4J$ = 1.6, 2H, C*H*-Cp), 6.51 (dt, $^3J$ = 5.2, $^4J$ = 1.6, 2H, C*H*-Cp), 6.43 (dt, $^3J$ = 5.2, $^4J$ = 1.6, 2H, C*H*-Cp), 6.16 (dt, $^3J$ = 5.2, $^4J$ = 1.6, 2H, C*H*-Cp), 2.50 (s, 6H, C*H$_3$*). $^{13}$C NMR (CDCl$_3$, 125 MHz, 25 °C): δ 149.0, 143.8, 142.0 (*C*q), 132.1, 131.9 (CH-Cp), 129.0 (CH-Ar), 123.7, 121.2 (CH-Cp), 22.6 (*CH$_3$*). ESI-MS (*m/z*): 259.15 ([M + H]$^+$), 258.15 ([M]). Anal. calcd. for C$_{20}$H$_{18}$ (258.36): C 92.98, H 7.02; found: C 93.27, H 6.53.

### 3.5. 1,4-Bis(cyclopenta-2,4-dien-1-ylidenemethyl)benzene (**1b**)

Using a protocol similar to that described above for **1a**, **1b** was prepared from cyclopentadiene (30.7 mL, 373.0 mmol), 1,3-terephthalaldehyde (10.0 g, 74.5 mmol), and pyrrolidine (9.3 mL, 112.0 mmol), and isolated as an orange powder (13.03 g, 56.7 mmol, 76%). $^1$H NMR (CDCl$_3$, 400 MHz, 25 °C): δ 7.63 (s, 4H, C*H*-Ar), 7.20 (s, 2H, C*H*-methine), 6.69 (m, 4H, C*H*-Cp), 6.52 (d, $^3J$ = 5.0, 2H, C*H*-Cp), 6.32 (d, $^3J$ = 5.0, 2H, C*H*-Cp). $^{13}$C NMR (CDCl$_3$, 125 MHz, 25 °C): δ 146.2, 137.6, (*C*q), 137.2, 136.1 (CH-Cp), 131.5 (=CH), 131.1 (CH-Ph), 127.4, 120.3 (CH-Cp). Anal. calcd. for C$_{18}$H$_{14}$ (230.30): C 93.87, H 6.13; found: C 93.98, H 6.56.

### 3.6. 1,3-Bis(1-(cyclopenta-2,4-dien-1-ylidene)ethyl)benzene (**1c**)

Using a protocol similar to that described above for **1a**, **1c** was prepared from cyclopentadiene (30.0 mL, 363.0 mmol), 1,3-diacetylbenzene (11.0 g, 68.0 mmol), and pyrrolidine (17.0 mL, 204.0 mmol), and isolated as an orange powder (14.9 g, 51.0 mmol, 85%). $^1$H NMR (CDCl$_3$, 400 MHz, 25 °C): δ 7.41 (broad m, 4H, C*H*-Ar), 6.66 (dt, $^3J$ = 5.2, $^4J$ = 1.8, 2H, C*H*-Cp), 6.60 (dt, $^3J$ = 5.2, $^4J$ = 1.6, 2H, C*H*-Cp), 6.51 (dt, $^3J$ = 5.2, $^4J$ = 1.6, 2H, C*H*-Cp), 6.21 (dt, $^3J$ = 5.2, $^4J$ = 1.6, 2H, C*H*-Cp), 2.58 (s, 6H, C*H$_3$*). $^{13}$C NMR (CDCl$_3$, 125 MHz, 25 °C): δ 149.2, 143.8, 141.9 (*C*q), 132.2, 131.9 (CH-Cp), 129.8, 129.2, 127.6 (CH-Ar), 123.6, 121.2 (CH-Cp), 22.7 (*CH$_3$*). ESI-MS (*m/z*): 259.15 ([M + H]$^+$). Anal. calcd. for C$_{20}$H$_{18}$ (258.36): C 92.98, H 7.02; found: C 93.02, H 7.13.

### 3.7. 1,4-Ph((Me)C-(3,6-tBu$_2$FluH)(CpH))$_2$ (**2a**)

In a Schlenk flask, to a solution of 3,6-di-*tert*-butyl-fluorene (2.17 g, 7.8 mmol) in THF (100 mL), was added *n*-BuLi (3.13 mL of a 2.5 M solution in hexane, 7.8 mmol). This solution was added dropwise to a solution of **1a** (1.00 g, 3.9 mmol) in THF (100 mL) at −10 °C over 10 min. After completion of the addition, the reaction mixture was stirred for 2 days at room temperature. The mixture was hydrolyzed with 10% aqueous hydrochloric acid (20 mL), the organic phase was dried over sodium sulfate, and volatiles were evaporated in vacuo. The solid residues was washed with pentane (200 mL) and dried under reduced in vacuo to afford a white powder (1.96 g, 2.4 mmol, 62%). $^1$H NMR (CDCl$_3$, 400 MHz, 25 °C) (mixtures of tautomers): δ 7.76 (m, 4H, C*H*-Ph), 7.66 (m, 4H, C*H*-Flu), 7.25–7.08 (m, 4H, C*H*-Flu), 7.01 (m, 1H, C*H*-Flu), 6.87 (t, $^3J$ = 9.0, 1H, C*H*-Flu), 6.62 (t, $^3J$ = 9.0, 1H, C*H*-Flu), 6.54–6.41 (m, 6H, C*H*-Flu + C*H*-Cp), 6.18 (m, 1H, C*H*-Cp), 4.96 (s, 2H, *H*-Flu), 3.19–2.99 (m, 4H, C*H$_2$*-Cp), 1.38 (m, 36H, C*H$_3$*-$^t$Bu), 1.12–1.07 (m, 6H, C*H$_3$*-bridge). $^{13}$C NMR (CDCl$_3$, 125 MHz, 25 °C): δ 156.4, 156.3, 156.2 (*C*q-Cbridge), 153.8 (*C*q), 150.2, 150.2, 150.1, 150.1 (*C*q-*C*-$^t$Bu), 145.5, 145.4 (*C*q), 142.8, 142.8, 142.6, 142.5, 142.3, 142.2 (*C*q), 134.5, 134.4, 134.3 (CH-Flu), 133.9, 133.8 (CH-Flu), 132.2, 131.9 (CH-Cp), 127.8, 127.8, 127.7, 127.5 (CH-Cp), 126.3 (CH-Cp), 125.7, 125.6, 125.5 (CH-Cp), 123.7, 123.6, 123.5, 123.4 (CH-Flu), 115.9, 115.9 (CH-Ph), 68.1 (*C*q), 55.4, 55.4, 55.3 (CH), 54.0, 53.9 (CH), 47.5, 47.4 (*C*q), 46.4, 46.3 (*C*q), 42.1, 41.0 (CH$_2$-Cp), 34.9, 34.8 (*C*q), 31.8 (*CH$_3$*-$^t$Bu). ESI-MS (*m/z*): 815.55 ([M + H]$^+$), 814.54 ([M]). Anal. calcd. for C$_{62}$H$_{70}$ (815.22): C 91.35, H 8.65; found: C 91.68, H 8.78.

### 3.8. 1,4-Ph((H)C-(3,6-tBu$_2$FluH)(CpH))$_2$ (**2b**)

Using a protocol similar to that described above for **2a**, compound **2b** was prepared from 3,6-di-*tert*-butyl-fluorene (4.83 g, 17.4 mmol), *n*-BuLi (7.0 mL of a 2.5 M solution in hexane, 17.4 mmol), **1b** (2.00 g, 8.7 mmol), and isolated as a white powder (4.12 g, 5.2 mmol, 60%). $^1$H NMR (CDCl$_3$, 400 MHz, 25 °C) (mixture of tautomers): δ 7.72 (m, 4H, C*H*-Ar), 7.10 (m, 8H, C*H*-Cp), 7.02 (m, 1H, C*H*-Cp), 6.88 (m, 1H, C*H*-Cp), 6.7 (m, 3H, C*H*-Cp), 6.45 (m, 2H, C*H*-Cp), 6.33 (m, 1H, C*H*-Cp), 6.18 (m, 1H, C*H*-Cp), 5.96 (m, 1H, C*H*-Cp), 4.52 (m, 2H, C*H*-Flu), 4.03 (m, 2H, C*H*-bridge), 2.96 (m, 3H, C*H*$_2$-Cp), 1.36 (s, 36H, C*H*$_3$-$^t$Bu). $^{13}$C NMR (CDCl$_3$, 125 MHz, 25 °C): δ 150.87, 150.69, 150.25, 150.21, 150.19, 150.17, 150.15, 150.13, 150.11, 150.09, 150.05 (Cq-C-$^t$Bu), 148.39, 148.34 (*Cq*), 143.57, 143.55, 143.44, 143.40, 143.37, 143.32 (*Cq*), 141.69, 141.65, 141.48, 141.45, 141.42, 141.39 (*Cq*), 140.67 (*Cq*), 134.50, 134.47, 134.41 (CH-Flu), 133.89, 133.81 (CH-Flu), 132.42, 131.43, 131.40, 131.37 (CH-Cp), 129.25, 129.22, 129.19, 129.16, 128.96 (CH-Cp), 128.90, 128.87, 128.82, 128.80, 128.72, 128.67, 128.62 (CH-Cp), 125.45, 125.33, 125.31, 125.28, 125.23, 125.19, 125.16 (CH-Cp), 123.64, 123.61, 123.57, 123.55, 123.51, 123.46, 123.41 (CH-Flu), 116.12, 116.07 (CH-Ph), 51.50, 51.46, 51.44, 51.22, 51.14, 51.12 (CH-bridge), 50.44, 50.39, 50.30, 50.20, 50.16 (*CH*), 43.15, 43.12, 41.14, 41.12 (CH$_2$-Cp), 34.90, 34.88 (*Cq*), 31.80 (CH$_3$-$^t$Bu). Mp: 240 °C. ESI-MS (*m/z*): 787.52 ([M + H]$^+$), 786.51 ([M]). Anal. calcd. for C$_{60}$H$_{66}$ (787.17): C 91.55, H 8.45; found: C 91.68, H 8.90.

### 3.9. 1,3-Ph(MeC-(3,6-tBu$_2$FluH)(CpH))$_2$ (**2c**)

Using a protocol similar to that described above for **2a**, compound **2c** was prepared from 3,6-di-*tert*-butyl-fluorene (2.17 g, 7.8 mmol), *n*-BuLi (3.13 mL of a 2.5 M solution in hexane, 7.8 mmol), and **1c** (1.00 g, 3.9 mmol). The final product **2c** was isolated as a white powder (469 mg, 0.58 mmol, 15%). $^1$H NMR (CDCl$_3$, 500 MHz, 25 °C): δ 7.80 (m, 1H, C*H*-Flu), 7.72–7.64 (m, 4H, C*H*-Ph), 7.53–7.30 (m, 4H, C*H*-Flu), 7.11 (m, 2H, C*H*-Flu), 7.04–6.99 (m, 1H, C*H*-Flu), 6.92–6.70 (m, 4H, C*H*-Flu), 6.49–6.34 (m, 4H, C*H*-Cp), 6.26 (m, 1H, C*H*-Cp), 6.13–6.07 (m, 1H, C*H*-Cp), 4.84 (m, 2H, C*H*-Flu), 3.01 (m, 4H, C*H*$_2$-Cp), 1.36–1.30 (s, 36H, C*H*$_3$-$^t$Bu), 1.19–1.06 (m, 6H, C*H*$_3$-bridge). $^{13}$C NMR (CDCl$_3$, 125 MHz, 25 °C): δ 156.3, 156.2 (Cq-Cbridge), 153.9 (*Cq*), 150.2, 150.1, 150.0, 149.9 (Cq-C-$^t$Bu), 142.8, 142.4, 142.1, 142.1 (*Cq*), 134.5, 134.4, 134.3 (CH-Flu), 133.8, 133.6 (CH-Flu), 132.0, 131.9 (CH-Cp), 128.0, 127.9, 127.8 (*CH*-Cp), 126.4, 126.3 (CH-Cp), 125.9, 125.8, 125.7, 125.6, 125.5 (CH-Cp), 123.6, 123.6, 123.5, 123.4, 123.4 (CH-Flu), 115.9, 115.9, 115.8 (CH-Ph), 66.0 (*Cq*), 55.5, 55.4 (*CH*), 54.2, 54.1 (*CH*), 48.0, 47.9 (*Cq*), 47.1, 47.0 (*Cq*), 42.0, 41.9, 41.0 (CH$_2$-Cp), 34.9, 34.8 (*Cq*), 31.7 (CH$_3$-$^t$Bu). ESI-MS (*m/z*): 853.51 ([M + K]$^+$), 837.54 ([M + Na]$^+$), 815.55 ([M + H]$^+$). Anal. calcd. for C$_{62}$H$_{70}$ (815.22): C 91.35, H 8.65; found: C 91.55, H 8.69.

### 3.10. PhMeC-(3,6-tBu$_2$FluH)(CpH) (**2a′**)

Using a protocol similar to that described above for **2a**, compound **2a′** was prepared from 3,6-di-*tert*-butyl-fluorene (0.97 g, 3.5 mmol), *n*-BuLi (1.4 mL of a 2.5 M solution in hexane, 3.5 mmol), and (1-(cyclopenta-2,4-dien-1-ylidene)ethyl)benzene (0.59 g, 3.5 mmol), and isolated as a white powder (0.65 g, 1.4 mmol, 40%). $^1$H NMR (CDCl$_3$, 500 MHz, 25 °C): δ 7.75 (dd, *J* = 7.1, 2.0, 1H, C*H*-Flu), 7.72 (t, *J* = 2.2, 1H, C*H*-Flu), 7.63 (d, *J* = 7.8, 2H, C*H*-Ph), 7.39 (td, *J* = 7.5, 4.4, 2H, C*H*-Ph), 7.30 (tt, *J* = 7.2, 1.4, 1H, C*H*), 7.14 (ddd, *J* = 13.8, 8.1, 2.0, 1H, C*H*-Flu), 7.00–6.88 (m, 2H, C*H*-Flu), 6.71 (d, *J* = 8.2, 1H, C*H*-Cp), 6.55 6.48 (m, 2H, C*H*-Flu), 6.39 (dd, *J* = 5.3, 1.5, 1H, C*H*-Cp), 6.24–6.18 (m, 1H, C*H*-Cp), 6.00 (dd, *J* = 13.9, 8.1, 1H, C*H*-Cp), 4.91 (d, *J* = 3.5, 1H, C*H*-Flu), 3.06 (dd, *J* = 7.1, 1.6, 1H, C*H*$_2$-Cp), 1.45–1.29 (m, 20H, C*H*$_3$-$^t$Bu), 1.07 (m, 3H, C*H*$_3$-bridge). $^{13}$C NMR (CDCl$_3$, 125 MHz, 25 °C): δ 156.04 (Cq), 153.70 (Cq), 150.20, 150.14, 150.08, 150.06 (Cq), 147.74, 147.17 (Cq), 142.84, 142.79, 142.34, 142.31, 142.20, 142.19, 142.15, 142.08 (Cq), 134.18, 134.01 (CH), 132.07, 131.96 (CH), 128.38 (CH), 127.89, 127.65, 127.57 (CH), 126.47, 126.32, 126.27 (CH), 125.66, 125.47, 125.45, 125.31 (CH), 123.67, 123.61, 123.54, 123.45 (CH), 115.93, 115.87, 115.81, 115.79 (CH), 55.56, 54.30 (CH-Flu), 47.73, 46.59, 42.01, 40.98 (CH$_2$-Cp), 34.89, 34.87, 34.81, 31.93, 31.87, 31.77, 31.72, 31.62 (CH$_3$-$^t$Bu), 18.48, 18.40 (CH$_3$). ESI-MS (*m/z*): 447.30 ([M +

H]$^+$), 277.19 ([3,6-di-*tert*-butyl-fluorene]$^+$). Anal. calcd. for C$_{34}$H$_{38}$ (446.67): C 91.42, H 8.58; found: C 91.54, H 8.71.

### 3.11. Ph(H)C-(3,6-tBu$_2$FluH)(CpH) (2b')

Using a protocol similar to that described above for **2a**, compound **2b'** was prepared from 3,6-di-*tert*-butyl-fluorene (4.70 g, 16.9 mmol), *n*-BuLi (6.8 mL of a 2.5 M solution in hexane, 16.9 mmol), and (cyclopenta-2,4-dien-1-ylidenemethyl)benzene (2.60 g, 16.9 mmol), and isolated as a white powder (0.70 g, 1.62 mmol, 10%). $^1$H NMR (CDCl$_3$, 400 MHz, 25 °C): δ 7.71 (m, 2H, C*H*-Flu), 7.24–7.12 (m, 6H, C*H*-Ph + C*H*-Flu), 7.04 (m, 1H, C*H*-Flu), 6.89 (d, *J* = 8.1, 1H, C*H*-Flu), 6.68–6.01 (m, 4H, C*H*-Cp), 4.54 (m, 1H, C*H*-*bridge*), 4.06 (m, 1H, C*H*-Flu), 3.12–2.89 (m, 2H, C*H*$_2$-Cp), 1.39–1.32 (m, 18H, C*H*$_3$-*t*Bu). $^{13}$C NMR (CDCl$_3$, 125 MHz, 25 °C): 150.72, 150.22, 150.15, 150.09 (*C*q), 148.38 (*C*q), 143.51, 143.22, 142.76 (*C*q), 141.45, 141.39 (*C*q), 134.34, 134.04 (*C*H), 132.42, 131.47 (*C*H), 129.28, 129.09, 129.04, 128.64, 128.54, 128.28, 128.21 (*C*H), 126.52, 125.34, 125.28, 125.20, 125.13 (*C*H), 123.68, 123.62, 123.56 (*C*H), 116.13, 116.09, 116.06 (*C*H), 51.54, 51.48, 50.61, 50.40 (*C*H), 43.16, 41.18 (*C*H$_2$-Cp), 34.91, 34.88, 31.88, 31.78, 31.76 (*C*H$_3$-*t*Bu). ESI-MS (*m/z*): 433.29 ([M + H]$^+$). Anal. calcd. for C$_{33}$H$_{36}$ (432.64): C 91.61, H 8.39; found: C 91.92, H 8.84.

### 3.12. 1,4-Ph{MeC-(3,6-tBu$_2$Flu)(Cp)ZrCl$_2$}$_2$ (3a-Zr)

To a solution of **2a** (0.500 g, 0.61 mmol) in diethyl ether (50 mL) was added under stirring *n*-BuLi (0.98 mL of a 2.0 M solution in hexane, 2.45 mmol). The solution was kept overnight at room temperature. ZrCl$_4$ (0.286 g, 1.23 mmol) was added to the reaction mixture with a bent finger. The resulting red mixture was stirred at room temperature overnight. Then, the mixture was evaporated in vacuo and CH$_2$Cl$_2$ (20 mL) was added. The resulting solution was filtered, and volatiles were evaporated in vacuo to give a red powder (0.528 g, 0.46 mmol, 76%). $^1$H NMR (CD$_2$Cl$_2$, 500 MHz, 25 °C): δ 8.20–8.10 (m, 3H, C*H*-Flu), 8.01–7.88 (4H, C*H*-Flu), 7.87–7.66 (4H, C*H*-Flu + C*H*-Ph), 7.58–7.49 (2H, C*H*-Ph), 7.40–7.06 (4H, 2H, C*H*-Flu), 6.63–6.56 (1H, C*H*-Cp), 6.46–6.28 (4H, C*H*-Cp), 6.00–5.77 (3H, C*H*-Cp), 2.68–2.57 (m, 6H, C*H*$_3$), 1.52–1.43 (d, 36H, C*H*$_3$-*t*Bu). $^{13}$C NMR (CD$_2$Cl$_2$, 125 MHz, 25 °C): δ 158.3 (*C*q-*t*Bu), 150.3 (*C*q-*t*Bu), 149.7 (*C*q-Ph), 145.3 (*C*q-Flu), 129.6–128.8 (*C*H-Ph), 128.1 (*C*q-Flu), 127.1 (*C*H-Flu), 126.8 (*C*H-Flu), 126.2 (*C*q-Flu), 126.4 (*C*H-Flu), 125.8 (*C*H-Flu), 123.2 (*C*H-Cp), 123.1 (*C*H-Cp), 122.7 (*C*H-Cp), 120.3 (*C*H-Cp), 119.8 (*C*H-Cp), 117.5 (*C*H-Cp), 112.4 (*C*q-Cp), 103.7–103.5 (*C*H-Cp), 102.5 (*C*H-Cp), 102.4 (*C*H-Cp), 77.4 (*C*-Flu), 53.8–53.4 (*C*H$_3$-bridge), 31.7–30.8 (*C*H$_3$-*t*Bu). APPI-MS (*m/z*): 1030.20 ([M]$^+$), 970.36 ([M–ZrCl$_2$]$^+$), 810.51 ([M–Zr$_2$Cl$_4$]$^+$). Anal. calc*d* for C$_{62}$H$_{66}$Cl$_4$Zr$_2$ (1135,45): C 65.58, H 5.86; found: C 65.42, H 6.00.

### 3.13. 1,4-Ph{MeC-(3,6-tBu$_2$Flu)(Cp)HfCl$_2$}$_2$ (3a-Hf)

Using a protocol similar to that described above for **3a-Zr**, compound **3a-Hf** was prepared from 1,4-bis(cyclopenta-2,4-dien-1-yl(3,6-di-*tert*-butyl-fluoren-9-yl)ethyl)benzene (0.500 g, 0.61 mmol), *n*-BuLi (0.98 mL of a 2.5 M solution in hexane, 2.45 mmol), and HfCl$_4$ (0.380 g, 1.23 mmol). The compound was recovered as an orange-yellow powder (0.520 g, 0.38 mmol, 62%). $^1$H NMR (toluene-*d*$_8$, 500 MHz, 25 °C): δ 8.33–8.10 (m, 4H, C*H*-Flu), 8.00 (s, 2H, C*H*-Flu), 7.77–7.23 (m, 7H, C*H*-Flu + C*H*-Ph), 6.79 (d, *J* = 9.2, 1H, C*H*-Flu), 6.34 (d, *J* = 9.3, 1H, C*H*-Flu), 6.17–5.85 (m, 4H, C*H*-Cp), 5.65–5.30 (m, 4H, C*H*-Cp), 2.30 (s, 6H, C*H*$_3$), 1.70–1.17 (m, 36H, C*H*$_3$-*t*Bu). $^{13}$C NMR (C$_6$D$_6$, 125 MHz, 25 °C): δ 150.02, 149.75, 149.71, 146.71, 132.18 (*C*q), 130.29, 129.93, 129.53, 129.40, 129.22, 128.76, 128.55, 128.45, 128.36, 128.25, 128.16, 128.06, 127.72, 127.43, 127.16, 126.79, 126.43, 125.25, 125.02, 123.49, 123.19, 122.97, 122.61, 122.47, 122.43, 122.34, 121.41, 121.20, 120.39, 120.16, 119.88, 119.74, 119.34, 116.98, 116.79, 115.48, 114.77, 101.70, 101.51, 100.27, 79.57, 78.89 (*C*-Flu), 49.95, 49.64 (*C*H$_3$), 35.46, 35.42, 35.37, 35.32, 35.27, 35.11, 32.21, 32.18, 32.05, 32.02, 32.00, 31.97, 31.96, 31.47, 29.92, 29.78, 27.45 (*C*H$_3$-*t*Bu). APPI-MS (*m/z*): 1310.3046 ([M]$^{+•}$), 1062.4287 ([M–HfCl$_2$]$^+$). Anal. calc*d* for C$_{62}$H$_{66}$Cl$_4$Hf$_2$ (1310.2850): C 56.85, H 5.08; found: C 56.73, H 5.16.

### 3.14. 1,4-Ph{(H)C-(3,6-tBu₂Flu)(Cp)ZrCl₂}₂ (**3b-Zr**)

Using a protocol similar to that described above for **3a-Zr**, compound **3b-Zr** was prepared from **2b** (0.660 g, 0.84 mmol), *n*-BuLi (1.37 mL of a 2.0 M solution in hexane, 3.37 mmol), and ZrCl₄ (0.392 g, 1.68 mmol). The compound was isolated as a red powder (0.350 g, 0.32 mmol, 38%). $^{1}$H NMR (CD₂Cl₂, 500 MHz, 25 °C): δ 8.16 (d, $^{3}J$ = 4.0, 2H, C*H*-Flu), 7.98 (s, 4H, C*H*-Ph), 7.67 (m, 4H, C*H*-Flu), 7.27 (dd, $^{3}J$ = 9.0, $^{4}J$ = 1.6, 2H, C*H*-Flu), 6.69 (d, $^{3}J$ = 9.0, 2H, C*H*-Flu), 6.64 (s, 2H, CH-*ansa*), 6.36 (dq, $^{3}J$ = 2.5, 4H, C*H*-Cp), 5.84 (dq, $^{3}J$ = 2.5, 4H, C*H*-Cp), 1.47 (d, 36H, C*H₃*-$^{t}$Bu). $^{13}$C NMR (CD₂Cl₂, 125 MHz, 25 °C): δ 150.4 (C-Flu-$^{t}$Bu), 137.6 (*C*q), 128.2 (CH-Ph), 127.8 (CH-Ph), 127.5 (*C*H-Flu), 122.8 (*C*q), 121.9 (*C*H-Flu), 120.3 (*C*q), 119.6 (*C*H-Flu), 119.6 (*C*H-Cp), 119.5 (*C*H-Cp), 117.6 (*C*H-Cp), 106.2 (*C*q), 104.9 (*C*H-Cp), 101.9 (*C*H-Cp), 73.9 (*C*-Flu), 41.7 (*C*H-bridge), 31.2 (*C*H₃-tBu). APPI-MS (*m/z*): 1102.1766 ([M]$^{+•}$), 942.3300 ([M–ZrCl₂]$^{+}$), 782.8 ([M–Zr₂Cl₄]$^{+}$). Anal. calcd. for C₆₀H₆₂Cl₄Zr₂ (1102.16997): C 65.08, H 5.64; found: C 65.75, H 6.01.

### 3.15. 1,3-Ph{MeC-(3,6-tBu₂Flu)(Cp)ZrCl₂}₂ (**3c-Zr**)

Using a protocol similar to that described above for **3a-Zr**, compound **3c-Zr** was prepared from **2c** (0.520 g, 0.64 mmol), *n*-BuLi (1.0 mL of a 2.5 M solution in hexane, 2.55 mmol), and ZrCl₄ (0.300 g, 1.27 mmol). The product was isolated as a red powder (0.630 g, 0.56 mmol, 87%). $^{1}$H NMR (C₆D₆, 500 MHz, 25 °C): δ 8.43–8.23 (m, 4H, C*H*-Flu), 8.05 (m, 2H, C*H*-Flu), 7.70 (dt, *J* = 7.8, 1.4, 1H, C*H*-Flu), 7.63–7.25 (m, 6H, C*H*-Flu + C*H*-Ph), 7.06–6.90 (m, 2H, C*H*-Ph), 6.47 (d, *J* = 9.1, 1H, C*H*-Ph), 6.28–6.23 (m, 1H, C*H*-Flu), 6.23–6.15 (m, 1H, C*H*-Cp), 6.13–6.04 (m, 1H, C*H*-Cp), 5.90–5.84 (m, 1H, C*H*-Cp), 5.74 (m, 1H, C*H*-Cp), 5.67–5.57 (m, 2H, C*H*-Cp), 5.55–5.36 (m, 1H, C*H*-Cp), 5.14 (d, *J* = 2.7, 1H, C*H*-Cp), 2.55–2.13 (m, 6H, C*H₃*), 1.51–1.29 (m, 36H, C*H₃*-*t*Bu). $^{13}$C NMR (C₆D₆, 125 MHz, 25 °C): δ 150.36, 150.10, 150.00, 149.79, 148.63, 148.55 (Cq), 131.99, 130.20, 129.34, 129.25, 129.19, 127.55, 127.27, 127.05, 126.53, 124.83, 124.53, 124.50, 123.91, 123.86, 123.84, 123.60, 123.52, 123.27, 123.22, 123.19, 123.00, 122.88, 122.85, 122.77, 122.60, 122.56, 120.56, 120.31, 120.21, 120.13, 120.10, 120.05, 119.90, 119.73, 117.86, 117.77, 117.30, 112.98, 112.63, 112.46, 104.92, 103.75, 103.57, 102.37, 101.31, 101.26, 79.19, 78.67, 78.63 (*C*-Flu), 50.49, 50.18, 50.09 (*C*H₃-bridge), 35.22, 35.18, 35.16, 35.08, 32.00, 31.97, 31.95, 31.92, 31.85, 31.78, 31.73, 31.70, 31.36, 30.95, 29.73 (*C*H₃-*t*Bu), 18.14. APPI-MS (*m/z*): 1130.2240. Anal. calc*d* for C₆₂H₆₆Cl₄Zr₂ (1130.2013): C 65.53, H 5.94; found: C 66.03, H 6.64.

### 3.16. {Ph(Me)C-(3,6-tBu₂Flu)(Cp)}ZrCl₂ (**3a'-Zr**)

Using a protocol similar to that described above for **3a-Zr**, compound **3b-Zr** was prepared from **2a'** (0.400 g 0.89 mmol), *n*-BuLi (0.72 mL of a 2.5 M solution in hexane, 1.79 mmol), and ZrCl₄ (0.209 g, 0.89 mmol). The compound was isolated as a red powder (0.410 g, 0.67 mmol, 76%). $^{1}$H NMR (toluene-*d₈*, 500 MHz, 25 °C): δ 8.07 (ddd, *J* = 7.0, 1.9, 0.7, 2H, C*H*-Flu), 7.43 (dt, *J* = 7.8, 1.7, 1H, C*H*-Ph), 7.39 (dd, *J* = 9.3, 0.8, 1H, C*H*-Flu), 7.24 (dt, *J* = 7.8, 1.7, 1H, C*H*-Ph), 7.22–7.16 (m, 2H, C*H*-Ph), 7.08–6.96 (m, 3H, C*H*-Flu), 6.88 (m, 1H, C*H*-Cp), 6.65 (m, 1H, C*H*-Cp), 6.12–5.76 (m, 2H, C*H*-Cp), 5.32 (t, *J* = 2.7, 2H), 2.01 (s, 3H, C*H₃*-bridge), 1.25 (m, 18H, C*H₃*-*t*Bu). $^{13}$C NMR (toluene-*d₈*, 125 MHz, 25 °C): δ 150.05 (C-Flu-*t*Bu), 149.74 (C-Flu-*t*Bu), 147.28 (Cq-Ph), 129.45, 129.19 (CH-Ph), 128.25 (CH-Flu), 127.29 (CH-Ph), 127.23 (CH-Flu), 125.72 (CH-Cp), 125.41 (CH-Cp), 123.65, 123.55, 123.25 (CH-Cp), 122.93, 122.68 (CH-Cp), 120.25 (CH-Cp), 119.94, 119.39, 117.46 (CH-Flu), 112.95 (Cq-Cp), 103.58, 102.30 (CH-Cp), 78.99 (*C*-Flu), 49.94 (Cq-*t*Bu), 35.15, 35.10 (Cq-*t*Bu), 31.77, 31.07 (*C*H₃-*t*Bu). APPI-MS (*m/z*): 604.1239 ([M]$^{+•}$), 444.2664 ([M–ZrCl₂]$^{+}$. Anal. calc*d* for C₃₄H₃₆Cl₂Zr (604.1241): C 67.30, H 5.98; found: C 67.82, H 6.09.

### 3.17. {Ph(H)C-(3,6-tBu₂Flu)(Cp)}ZrCl₂ (**3b'-Zr**)

Using a protocol similar to that described above for **3a-Zr**, compound **3b'-Zr** was prepared from **2b'** (0.430 g, 0.99 mmol), *n*-BuLi (0.81 mL of a 2.5 M solution in hexane, 1.99 mmol), and ZrCl₄ (0.230 g, 0.99 mmol). The product was isolated as a red powder (0.540 g, 0.86 mmol, 87%). $^{1}$H NMR (C₆D₆,

500 MHz, 25 °C): δ 8.24 (s, 2H, C*H*-Flu), 7.55 (d, *J* = 7.4, 2H, C*H*-Ph), 7.42 (d, *J* = 9.0, 1H, C*H*-Ph), 7.24 (m, 3H, C*H*-Ph), 6.99 (d, *J* = 8.8, 1H, C*H*-Flu), 6.92 (d, *J* = 9.1, 1H, C*H*-Flu), 6.51 (d, *J* = 9.0, 1H, C*H*-Flu), 6.17–6.05 (m, 2H, C*H*-Flu), 5.80 (s, 1H, C*H*), 5.48 (t, *J* = 2.8, 1H, C*H*-Cp), 5.29 (t, *J* = 2.8, 1H, C*H*-Cp), 1.39 (s, 18H, C*H*$_3$-$^t$Bu). $^{13}$C NMR (C$_6$D$_6$, 125 MHz, 25 °C): δ 150.08 (*C*-Flu-*t*Bu), 149.89 (C-Flu-*tBu*), 138.64 (C-Flu), 128.78, 128.38 (C-*t*Bu), 127.96, 127.21 (CH-Flu), 122.77, 122.31, 122.24, 121.54 (CH-Cp), 119.95, 119.76, 119.74, 119.51, 117.01 (CH-Flu), 106.86, 105.16, 101.87 (CH-Cp), 73.63 (*C*-Flu), 41.81 (*C*q), 34.98, 34.85 (*C*q), 31.57, 31.49 (*C*H$_3$-*t*Bu). APPI-MS (*m/z*): 590.1115 ([M]$^{+\bullet}$), 430.2659 ([M–ZrCl$_2$]$^+$). Anal. calcd for C$_{33}$H$_{34}$Cl$_2$Zr (590.1085): C 66.87, H 5.78; found: C 67.11, H 5.97.

### 3.18. {Ph(H)C-(3,6-tBu₂Flu)(Cp)}HfCl₂ (**3b'-Hf**)

This compound was prepared as described above for **3a** starting from **2b'** (0.090 g, 0.20 mmol), *n*-BuLi (0.16 mL of a 2.5 M solution in hexane, 0.41 mmol), and HfCl$_4$ (0.060 g, 0.2 mmol). The product was isolated as an orange powder (0.076 g, 0.86 mmol, 56%). $^1$H NMR (C$_6$D$_6$, 500 MHz, 25 °C): δ 8.22 (dt, *J* = 9.0, 1.9, 2H, C*H*-Flu), 7.56 (dt, *J* = 8.0, 1.4, 2H, C*H*-Flu), 7.40 (dd, *J* = 9.0, 1.7, 1H, C*H*-Ph), 7.28–7.20 (m, 3H, C*H*-Ph), 7.04 (d, *J* = 9.0, 1H), 6.91 (dd, *J* = 9.1, 1.8, 1H, C*H*-Ph), 6.57 (dd, *J* = 9.1, 1.8, 1H, C*H*-Ph), 6.09–6.00 (m, 2H, C*H*-Flu), 5.85 (s, 1H, C*H*-Cp), 5.44 (q, *J* = 2.7, 1H, C*H*-Cp), 5.25 (q, *J* = 2.8, 1H, C*H*-Cp), 1.39 (s, 18H, C*H*$_3$-$^t$Bu). $^{13}$C NMR (C$_6$D$_6$, 125 MHz, 25 °C): δ 149.80, 149.60 (*C*q), 139.28, 122.45, 121.64, 121.52, 120.32, 119.96, 119.92, 119.42, 118.78, 116.70, 116.43, 110.06, 103.15, 99.85, 74.22 (*C*-Flu), 42.10, 35.32, 35.18, 34.90, 34.84, 31.94, 31.85, 31.83, 31.79, 31.74, 31.70, 31.64, 31.52 (CH$_3$-*t*Bu). APPI-MS (*m/z*): 680.1497 ([M]$^{+\bullet}$), 430.2659 ([M–HfCl$_2$]$^+$), 278.2035 (C$_{21}$H$_{26}$, [3,6-di-*tert*-butyl-fluorene + H]$^{+\bullet}$). Anal. calc*d*. for C$_{33}$H$_{34}$Cl$_2$Hf (680.1503): C 58.29, H 5.04; found: C 58.55, H 5.26.

### 3.19. Ethylene Homopolymerization and Ethylene/1-hexene Copolymerization

Polymerization experiments were performed in a 300 mL high-pressure glass reactor equipped with a mechanical stirrer (Pelton turbine) and externally heated with a double mantle with a circulating water bath. The reactor was filled with toluene (100 mL), 1-hexene comonomer (when relevant; typically 2.5 mL), and MAO (0.20 mL of a 30 wt-% solution in toluene) and pressurized at 5.5 bar of ethylene (Air Liquide, 99.99%). The reactor was thermally equilibrated at the desired temperature for 30 min, the ethylene pressure was decreased to 1 bar, and a solution of the catalyst precursor in toluene (ca. 2 mL) was added by syringe. The ethylene pressure was immediately increased to 5.5 bar (kept constant with a back regulator), and the solution was stirred for the desired time (typically 15 min). The temperature inside the reactor (typically 60 °C) was monitored using a thermocouple. The polymerization was stopped by venting the vessel and quenching with a 10% HCl solution in methanol (ca. 2 mL). The polymer was precipitated in methanol (ca. 200 mL), and 35% aqueous HCl (ca. 1 mL) was added to dissolve possible catalyst residues. The polymer was collected by filtration, washed with methanol (ca. 200 mL), and dried under vacuum overnight.

### 3.20. Crystal Structure Determination of **3a'-Zr**

Diffraction data were collected at 150 K using a Bruker APEX CCD diffractometer (Bruker, Billerica, MA, USA) with graphite-monochromated MoKα radiation (λ = 0.71073 Å). A combination of ω- and φ-scans was carried out to obtain at least a unique data set. The crystal structures were solved by direct methods, and the remaining atoms were located from a difference Fourier synthesis followed by full-matrix least-squares refinement based on F2 (programs SIR97 [43] and SHELXL-97 [44]). Hydrogen atoms were placed at calculated positions and forced to ride on the attached atom. All non-hydrogen atoms were refined with anisotropic displacement parameters. The locations of the largest peaks in the final difference Fourier map calculation as well as the magnitude of the residual electron densities were of no chemical significance. The main crystal and refinement data are summarized in Table S1. Crystal data, details of data collection, and structure refinement for compound **3a'-Zr** (CCDC 1863222) can be obtained from the Cambridge Crystallographic Data Centre via www.ccdc.cam.ac.uk/data_request/cif.

*3.21. Computational Details*

All calculations were performed with the TURBOMOLE program package using density functional theory (DFT) [45–48]. The gradient-corrected density functional BP86 in combination with the resolution identity approximation (RI) [49,50] was applied for the geometry optimizations of a stationary point. The triple-$\zeta$ zeta valence quality basis set def-TZVP was used for all atoms [51].

The stationary points were characterized as energy minima (no negative Hessian eigenvalues) by vibrational frequency calculations at the same level of theory.

## 4. Conclusions

The synthesis of original bis(Cp/Flu) ligand systems linked at the C1-bridge through a phenylene group was developed starting from difulvene precursors. These ligand platforms were utilized for the preparation of homodinuclear zirconium and hafnium bis(dichloro *ansa*-metallocene) complexes via a regular salt-metathesis metallation protocol. The synthesis of a heterodinuclear zirconium/hafnium bis(dichloro *ansa*-metallocene) was also performed, although the desired product was generated as a statistical mixture with the corresponding homodinuclear complexes. For the first time, an advanced APPI mass-spectrometric method was applied to the characterization of dinuclear bis(ansa-metallocene) complexes and their mononuclear ansa-metallocene analogues, and relevant data were obtained.

Ethylene homopolymerization as well as ethylene/1-hexene copolymerization were conducted using the homodinuclear dichloro catalyst precursors, as well as with their mononuclear analogues, in combination with MAO. Limited cooperativity evidence has been observed with dinuclear systems so far, with, in some cases, slightly different molecular weights or a greater formation of short methyl and ethyl branches as compared to the mononuclear reference systems. The apparent lack of significant cooperative behavior observed for the dinuclear systems was substantiated by a computational analysis. Thus, the computed *para-* and *meta*-phenylene-bridged *neutral* dinuclear structures suggest that the two metallocenic fragments may orientate their coordination spheres in opposite directions, hence resulting in distant (>9 Å) isolated metal centers. Further investigations in our laboratories are focused on the elaboration of other polynuclear precatalysts with improved performances and identification of the nature of possible intermetallic cooperative effects.

**Supplementary Materials:** The following are available online at http://www.mdpi.com/2073-4344/8/11/558/s1, Figure S1: [1]H NMR spectrum of 1a; Figure S2: [13]C NMR spectrum of 1a; Figure S3: ASAP mass spectrum of 1a; Figure S4: [1]H NMR spectrum of 1b; Figure S5: [13]C NMR spectrum of 1b; Figure S6: [1]H NMR spectrum of 1c; Figure S7: [13]C NMR spectrum of 1c; Figure S8: ASAP mass spectrum of 1c; Figure S9: [1]H NMR spectrum of 2a; Figure S10: [13]C NMR spectrum of 2a; Figure S11: ASAP mass spectrum of 2a; Figure S12: [1]H NMR spectrum of 2b; Figure S13: [13]C NMR spectrum of 2b; Figure S14: ASAP mass spectrum of 2b; Figure S15: [1]H NMR spectrum of 2c; Figure S16: [13]C NMR spectrum of 2c; Figure S17: ASAP mass spectrum of 2c; Figure S18: [1]H NMR spectrum of 2a'; Figure S19: [13]C NMR spectrum of 2a'; Figure S20: ASAP mass spectrum of 2a'; Figure S21: [1]H NMR spectrum of 2b'; Figure S22: [13]C NMR spectrum of 2b'; Figure S23: ASAP mass spectrum of 2b'; Figure S24: [1]H NMR spectrum of 3a-Zr$_2$; Figure S25: HMBC spectrum of 3a-Zr$_2$; Figure S26: HSQC spectrum of 3a-Zr$_2$; Figure S27: APPI-IMMS of 3a-Zr$_2$; Figure S28: [1]H NMR spectrumof 3a-Hf$_2$; Figure S29: [13]C NMR spectrum of 3a-Hf$_2$; Figure S30: APPI-IMMS of 3a-Hf$_2$; Figure S31: [1]H NMR spectrum of 3b-Zr$_2$; Figure S32: HMBC spectrum of 3b-Zr$_2$; Figure S33: HSQC spectrum of 3b-Zr$_2$; Figure S34: APPI-IMMS of 3b-Zr$_2$; Figure S35: [1]H NMR spectrum of 3c-Zr$_2$; Figure S36: [13]C NMR spectrum of 3c-Zr$_2$; Figure S37: [1]H NMR spectrum of 3a'-Zr; Figure S38: [13]C NMR spectrum of 3a'-Zr; Figure S39: APPI-IMMS of 3a'-Zr; Figure S40: [1]H NMR spectrum of 3b'-Zr; Figure S41: [13]C NMR spectrum of 3b'-Zr; Figure S42: APPI-IMMS of 3b'-Zr; Figure S43: [1]H NMR spectrum of 3b'-Hf; Figure S44: [13]C NMR spectrum of 3b'-Hf; Figure S45: APPI-IMMS of 3b'-Hf; Figure S46: [13]C NMR spectrum of PE (Table 1, run 1); Figure S47: GPC trace of PE (Table 1, run 1) obtained with 3a-Zr$_2$; Figure S48: [13]C NMR spectrum of PE (Table 1, run 3); Figure S49: [13]C NMR spectrum of PE (Table 1, run 5); Figure S50: GPC trace of PE (Table 1, run 5) obtained with 3b-Zr$_2$; Figure S51: GPC trace of PE (Table 1, run 6) obtained with 3b-Zr$_2$; Figure S52: GPC trace of PE (Table 1, run 7) obtained with 3b'-Zr; Figure S53: GPC trace of PE (Table 1, run 8) obtained with 3b'-Zr; Figure S54. GPC trace of PE (Table 1, run 9) obtained with 3c-Zr$_2$; Figure S55: GPC trace of PE (Table 1, run 12) obtained with 3b'-Hf; Figure S57: GPC trace of PE/PHex (Table 2, run 1) obtained with 3a-Zr$_2$; Figure S58: [13]C NMR spectrum of PE/PHex (Table 2, run 4); Figure S59: [13]C NMR spectrum of PE/PHex (Table 2, run 5); Figure S60: GPC trace of PE/PHex (Table 2, run 5) obtained with 3b-Zr$_2$; Figure S61: GPC trace of PE/PHex (Table 2, run 6) obtained

with 3b-Zr$_2$; Figure S62. GPC trace of PE/PHex (Table 2, run 7) obtained with 3b'-Zr; Figure S63: GPC trace of PE/PHex (Table 2, run 8) obtained with 3b'-Zr; Figure S64: GPC trace of PE/PHex (Table 2, run 9) obtained with 3c-Zr$_2$; Figure S65: GPC trace of PE/PHex (Table 2, run 10) obtained with 3a-Hf$_2$; Figure S66: GPC trace of PE/PHex (Table 2, run 11) obtained with 3a-Hf$_2$; Figure S67: Molecular structure of 3a'-Zr; Table S1: Summary of Crystal and Refinement Data for Compound 3a'-Zr; Figure S68: DFT-optimized structures of $C_s$-symmetric and $C_1$-symmetric isomers of 3a-Zr$_2$; Figure S69: DFT-optimized structures of $C_s$-symmetric and $C_1$-symmetric isomers of 3c-Zr$_2$.

**Author Contributions:** G.S. (investigation), M.F. (investigation), L.B. (investigation), A.V. (conceptualization, project administration), A.W. (conceptualization), J.-M.B. (project administration), C.A. (data analysis), P.G. (data analysis), J.-F.C. (conceptualization, supervision, writing—review and editing), E.K. (conceptualization, investigation, supervision, writing—original draft preparation, writing—review and editing).

**Funding:** This research received no external funding.

**Acknowledgments:** This work was supported by Total S. A. and Total Research and Technologies Feluy and Gonfreville (postdoctoral and PhD grants, respectively, to G.S. and M.F.).

**Conflicts of Interest:** The authors declare no conflict of interest.

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
