# Peer review of "Synthesis, APPI Mass-Spectrometric Characterization, and Polymerization Studies of Group 4 Dinuclear Bis(ansa-metallocene) Complexes"

_catalysts, doi:10.3390/catal8110558_

Reviewer 1 Report

The authors report the preparation and characterization of Zr and Hf homodinuclear  bis(ansa-metallocene) complexes, which are tested, together with their monometallic analogs, in  ethylene hompolymerization and ethylene/1-hexene copolymerization. The manuscript is well written, clear and concise. The use of APPI spectrometry for characterization is significant. My concern is that there are no significant results...or better no advantage is reported when using dinuclear catalysts rather than mononuclear ones (despite the weak promise in the abstract). Apparently no inter metal coperative effects are operative, at least in catalysis. Any attempt to prepare the heterodinuclear compounds failed. Thus I think that the manuscript in the present form is not meaningful for the reader, unless we consider important to discourage this approach to get efficient catalysts. I wonder if the authors have considered to build some models to evaluate the inter metal distance in the isomers and recall some mechanistic hypothesis...well at least, if a catalyst doesn't work, an explanation should be provided to the reader and so also an unlucky attempt provides hints for rational catalysts'design.

 Author Response

Re: This is a pertinent remark. As suggested, we have indeed carried out a computational analysis for several putative structures of dinuclear complexes and the corresponding results have been included in the revised manuscript and Supporting Information. 

In the Discussion (p 5), the following paragraph has been added:
In order to get a better clue about possible structures of the dinuclear bis(metallocenes) the corresponding geometries of the two Cs- and Ci-symmetric isomers of 3a-Zr2(Fig. S68; see Experimental Section for details) and the two Cs- and C1-symmetric isomers of 3c-Zr2(Fig. S69) were modeled by DFT computations.  Noteworthy, the optimized geometries of both dinuclear systems 3a-Zr2and 3c-Zr2featured relatively long Zr¼Zr intermetallic distances of 10.510.8 and 9.29.8 Å, respectively.  Also, the respective orientations of the metallocenic fragments in these structures resulted in the coordination sites, represented by the chlorine ligands, pointing in opposite directions.  Such orientation of the metallocenic moieties in both para- and meta-phenylene-bridged systems may not be favorable to the mutual approach of the two metal centers in dinuclear active species derived thereof during polymerization.  Note, however, that the above observations were made on the most stable neutralisomers as determined by DFT and they do not necessarily reflect the proximity that can be reached from dynamic conformations in those species; also, the behavior of the active, cationicspecies, associated to counterionic moieties may be quite different.”
            Also, in the Conclusion (p 15), the following sentence has been added:

“The apparent lack of significant cooperative behavior observed for the dinuclear systems was substantiated by a computational analysis.  The computed para- and meta-phenylene-bridged neutraldinuclear structures suggest that the two metallocenic fragments may orientate their coordination spheres in opposite directions, hence resulting in distant (> 9 Å) isolated metal centers.”

Reviewer 2 Report

The ligand platforms p- or m-Ph{-CR(3,6-tBu2Fiu)(Cp)}2 were synthesized and characterized by NMR spectra, mass spectra, and elemental analysis. The corresponding homo-dinuclear and hetero-dinuclear bis(dichloro ansa-metallocene) complexes and mononuclear ansa-metallocene with Zr and Hf were prepared, and also characterized by NMR spectra, mass spectra, and elemental analysis. The synthesis and characterization are doing well down.

The cooperativity of dinuclear systems is similar (in some cases are slightly better) with mononuclear systems. We are looking forward the high catalytic activity dinuclear systems are created.

Author Response

Re : - no comments requested

Reviewer 3 Report

The authors reported the synthesis, APPI Mass-Spectrometric characterization, and polymerization studies of Group 4 dinuclear bis(ansa-metallocene) complexes. They examined and described the results in the text fully. I wish to recommend this manuscript to "Catalysts".

My opinions

1) What is the reason for the low yield of bis(fulveve) 1c?  Is compound 1c unstable?  Fulveves are not so stable compounds.

2) I recommend to note the synthesis of cyclopentadiene in General considerations (Materials and Methods).

Author Response

1) What is the reason for the low yield of bis(fulveve) 1c?  Is compound 1c unstable?  Fulveves are not so stable compounds.

Re: Bis(fulvene) 1cwas isolated in 85% yield and appeared to be stable under usual conditions.  On the other hand, a nucleophilic addition reaction between this meta-phenylene-bridged bis(fulvene) 1cand tBu2-fluorenyl-lithium (2 equiv) resulted in the corresponding proligand 2cin a lower yield (22 %) as compared to the para-phenylene-bridged analogues 2aand 2b.  We understand that this Reviewer pointed out the lower yiled of this product (2cand not 1c).  We rationalized this result due to a larger steric hindrance in the final meta-phenylene-bridged product imposed by the two very bulky {Cp/Flu} moieties, as compared to the paraanalogues.  This comment has been added in the revised manuscript, on p 3.  

 2) I recommend to note the synthesis of cyclopentadiene in General considerations (Materials and Methods).

 Re:  The commercial origin of cyclopentadiene has been added in the Materials and Methods section (p 10).

Round  2

Reviewer 1 Report

I have appreciated that modeling results have been introduced and support the experimental observations, within the limits clearly outlined by the authors in the text. In view also of the other reviwers'opinion, the paper now is suitable for publication. Minor recommendation: please add the xyz coordinates of the optimized structures in the ESI and not only the figures, to allow reproducibility of your results or simply easier inspection of the computed structures by the reader.